# Continuous generation of topological defects in a passively driven nematic liquid crystal

**Maruša Mur** [1], **Žiga Kos** [1,2], **Miha Ravnik** [1,2] **& Igor Mušević** [1,2] ✉

Synthetic active matter is emerging as the prime route for the realisation of biological mechanisms such as locomotion, active mixing, and self-organisation in soft materials. In particular, passive nematic complex fluids are known to form out-of-equilibrium states with topological defects, but their locomotion, activation and experimental realization has been developed and understood to only a limited extent. Here, we report that the concentration-driven flow of small molecules triggers turbulent flow in the thin film of a nematic liquid crystal that continuously generates pairs of topological defects with an integer topological charge. The diffusion results in the formation of counter-rotating vortex rolls in the liquid crystal, which above a velocity threshold transform into a turbulent flow with continuous generation and annihilation of the defect pairs. The pairs of defects are created by the self-amplifying splay instability between the vortices, until a pair of oppositely charged defects is formed.

Liquid crystals (LCs) are complex liquids that are rich in topological defects[1–3] and can be studied using optical methods. The topological aspects of LCs have been studied intensively in LC emulsions[4] and colloids[5,6], with beautiful realizations of complex topologies in nematic handle-bodies[7,8] and knotted colloidal nematics[9–12], testing the fundamental laws of topology. In these systems the topological defects are created by aligning the LC field along the closed surface of the inclusion, thereby generating the topological charges required by the Gauss-Bonnet theorem for vector fields. A variety of topological defects can be observed in chiral nematic droplets, where the spherical confinement of a chiral field generates a number of unit and higher topological charges[13–16].

Topological defects are ubiquitous in nature. They are the remnants of the symmetry breaking during the phase transition in matter and fields. They occur as vortices in superfluid helium[17,18] and Bose-Einstein condensates[19], Abrikosov vortices in type-II superconductors[20], and vortex-like structures in soft chiral ferromagnets[21]. In LCs the topological defects are regions where the orientational order parameter cannot be defined and the LC loses its orientation. The LC defects in three-spatial dimensions appear in the form of point defects called hedgehogs and their variants, closed defect loops. In two dimensions the defects are characterized by their winding number[22] and the head-tail nematic symmetry gives rise to defects with a half-integer winding number. In three-dimensional space the defects can have the shape of a point, a line emanating from and ending on a surface, or a closed defect loop. The defects in 3D are characterized by a topological charge that is determined by using the Gaussian integration of the flux of the orientational field through a closed surface embedding the defect. Hyperbolic and radial hedge-hogs carry a unit topological charge −1 and +1, respectively[1,4,5]. Higher charges have been observed in highly confined chiral nematics[23]. Topological charge is a conserved quantity and defects always appear in oppositely charged pairs. If point charges are attached to the surface, they are called boojums, exhibiting radial or hyperbolic configurations, with 2D topological charge $k = +1$ and $k = −1$, respectively. Topological defects and textures in nematics can be created by the rapid crossing of the isotropic-to-nematic phase transition, which is achieved by a quick pressure or temperature quench[24], thus testing the Kibble-Zurek mechanism of topological charge production. A rapid temperature quench with extremely fast cooling rates can be

[1]Condensed Matter Physics Department, J. Stefan Institute, Jamova 39, 1000 Ljubljana, Slovenia. [2]Faculty of Mathematics and Physics, University of Ljubljana, Jadranska 19, 1000 Ljubljana, Slovenia. ✉e-mail: igor.musevic@ijs.si

generated locally using optical tweezers, enabling controlled defect creation and manipulation[9,25].

The irregular-chaotic flow of LCs is capable of continuously driving spontaneous topological defect formation and annihilation. This phenomenon has recently received a lot of attention, both in theory[26–31] and experiments[32–35]. In active nematics the flow is perpetuated by the activity of the LC constituents themselves, whereas in passive nematics or smectics it is driven externally, i.e., by an external electric field[36–38]. The flow of a LC is always coupled with the nematic orientational order including the director field, which means that it can give rise to substantial local director re-orientations and can generate topological defects. A theoretical description of nematic hydrodynamics was developed by Ericksen[39], Leslie[40] and Parodi[41] and then generalized to tensorial order parameter formulation[42,43]. In shear flow the LC material can be flow-aligning, i.e., the director aligns at a fixed small angle to the flow velocity, or tumbling, i.e., the director does not reach a steady configuration, but experiences tumbling, rotating in the shear plane due to hydrodynamic torques[44]. Most nematic LCs are flow-aligning, the tumbling response is less usual and is often associated with less anisotropic nematic building blocks/molecules. LCs of similar composition can experience different behaviors: e.g., 5CB (4-pentyl-4′-cyanobiphenyl) is a flow-aligning nematic LC, whereas 8CB (4-octyl-4′-cyanobiphenyl) with a very similar molecular constitution appears to be tumbling almost throughout its nematic range. It transitions into the flow-aligning regime only close to the isotropic transition[45]. In this study we use the 8CB LC in its nematic phase.

Here we show that topological defects are spontaneously created and annihilated over a timescale of several minutes in a LC driven by an osmotic flow of small molecules that were added to the LC. The phenomenon of defect production is similar to that in active nematics[32,34], and is the result of the onset of chaotic flow in the LC. While the turbulence in active nematics is induced by the mechanical activity of individual constituents of the active nematic, our LC moves solely because of the flow of non-LC molecules, diffusing into the LC due to the concentration gradient. At first, the flow of the LC appears laminar, but after some time, counter-rotating vortex rolls are formed in the liquid crystal. Above a velocity threshold these rolls transform into a turbulent flow with continuous generation and annihilation of the defect pairs. The pairs of defects are created by the self-amplifying splay instability between the vortices, until a pair of oppositely charged defects is formed. The experiments are corroborated with numerical modeling of coupled material fields: nematic order, concentration of diffusing molecules and material flow, showing a very good agreement.

## Results

Our experiments were all performed at 25 °C and we use the 8CB (4-octyl-4′-cyanobiphenyl) LC that exhibits the smectic A phase at room temperature, has a phase transition to the nematic phase at 33 °C and becomes isotropic above 40 °C. We place a small drop of 8CB on a clean glass plate under a polarizing optical microscope and cover it with a large drop of water with 0.4 wt% of HMPP (2-hydroxy-2-methylpropiophenone photoinitiator). After a couple of minutes, the 8CB droplet suddenly and unexpectedly 'explodes' (see Supplementary Movie 1), and dramatically increases in diameter. A detailed explanation of this phenomenon is given in Supplementary Note 1. A wide, corona-like band of spontaneously flowing nematic LC appears around the motionless core of the LC drop, as shown in Fig. 1(a). The nematic corona-like band is clearly seen under crossed polarizers in Fig. 1(a), showing a very dynamic, flame-like fluid flow (see Supplementary Movie 2). The LC is vividly and continuously flowing in a chaotic manner and we are able to observe individual topological defects continually forming and annihilating. In bright-field images the topological defects appear as distinct dark spots and in cross-polarized images the defects are seen as sets of four dark brushes radiating out from a single point, which is the core of the defect. The defects are shown by the white arrows in the inset to Fig. 1(a).

We used confocal imaging to analyze the vertical cross-sections of the LC drop after the explosion. The water and LC were dyed with two different fluorescent dyes that provided a good color contrast. The 8CB LC was doped with Nile red fluorescent dye providing a red emission from the dyed LC. The water was dyed with Fluorescein-5-isothiocyate (FITC) fluorescent dye that emitted green-colored light, but is artificially colored cyan in Fig. 1(b) for the purpose of clarity. Figure 1(b) shows vertical cross-section of the LC drop (in red) in contact with the water-HMPP solution (in cyan). The fluorescent light from Nile red is seen only from a thin layer near the LC-glass interface. This is because the smectic phase is strongly scattering the excitation light and hence no fluorescence is excited deep in the smectic A part of the LC. We can clearly see that after the 'explosion' the LC is in contact with the outside air and the LC-water interface acquires a concave shape, as illustrated in the confocal cross-section in Fig. 1(b). Note that the LC-air interface is not visible in Fig. 1(b), it is only indicated with a dashed line. The core of the LC droplet remains fixed to the bottom glass plate, but from there the LC spreads along the water surface and forms a thin film on top of the water layer.

Because the surface anchoring of 8CB with air is homeotropic and it is planar at water interface, the thin nematic layer must be hybrid, as shown schematically in Fig. 1(c). The thickness profile of this hybrid nematic layer was reconstructed from the interference colors in a cross-polarized image in Fig. 1(a), and is shown in Fig. 1(d). We can see that the defects are only generated in the shaded part of the nematic film that is thinner than a few micrometers. It is important to note that in all the experiments we could only observe defects with four dark brushes emerging from a single point, which indicates an integer, |1| winding number of these defects. This is different than in the experimentally realized 2D active nematics[32,34] where only fractional |1/2| winding-number topological defects are formed. This is also different from the recently observed zero topological charge closed disclination loops in an active nematic driven by microtubule bundles[46]. The four dark brushes could either correspond to the surface defects, called boojums, or to bulk point defects, hedgehogs, which should be located inside the hybrid LC film. We are convinced that the observed defects are in fact surface boojums, residing at the LC-water interface, as schematically presented in Fig. 1(e). These kinds of defects were previously reported for a static 8CB hybrid nematic film between water and air[47,48].

The corona-like band of spontaneously flowing LC is in the nematic phase, although the experiments are done at room temperature where pure 8CB is in the smectic A phase. This is explained by difusion of HMPP photoinitiator from water to 8CB. The HMPP is poorly soluble in water, and it prefers to dissolve in 8CB. Therefore, when the water solution of HMPP comes into contact with the LC, the HMPP starts to diffuse from the water into the LC. After some HMPP is incorporated into the LC via molecular diffusion, it decreases the orientational order of the 8CB, and that causes a phase transformation of some parts of the smectic A droplet into the nematic phase at the same temperature. In Fig. 1(a) even whole droplet of 8CB has been transformed into the nematic phase due to high concentration of HMPP.

### Formation of topological defects

In the experiment, pairs of oppositely charged topological defects are therefore continuously created and annihilated in the thin hybrid nematic film for several minutes or even tens of minutes. Most of the defects are created at the isotropic edge in the thin part of the LC film and then move towards the thicker part of the film, i.e., towards the core of the LC drop, as shown in selected images in Figs. 2(a–f).

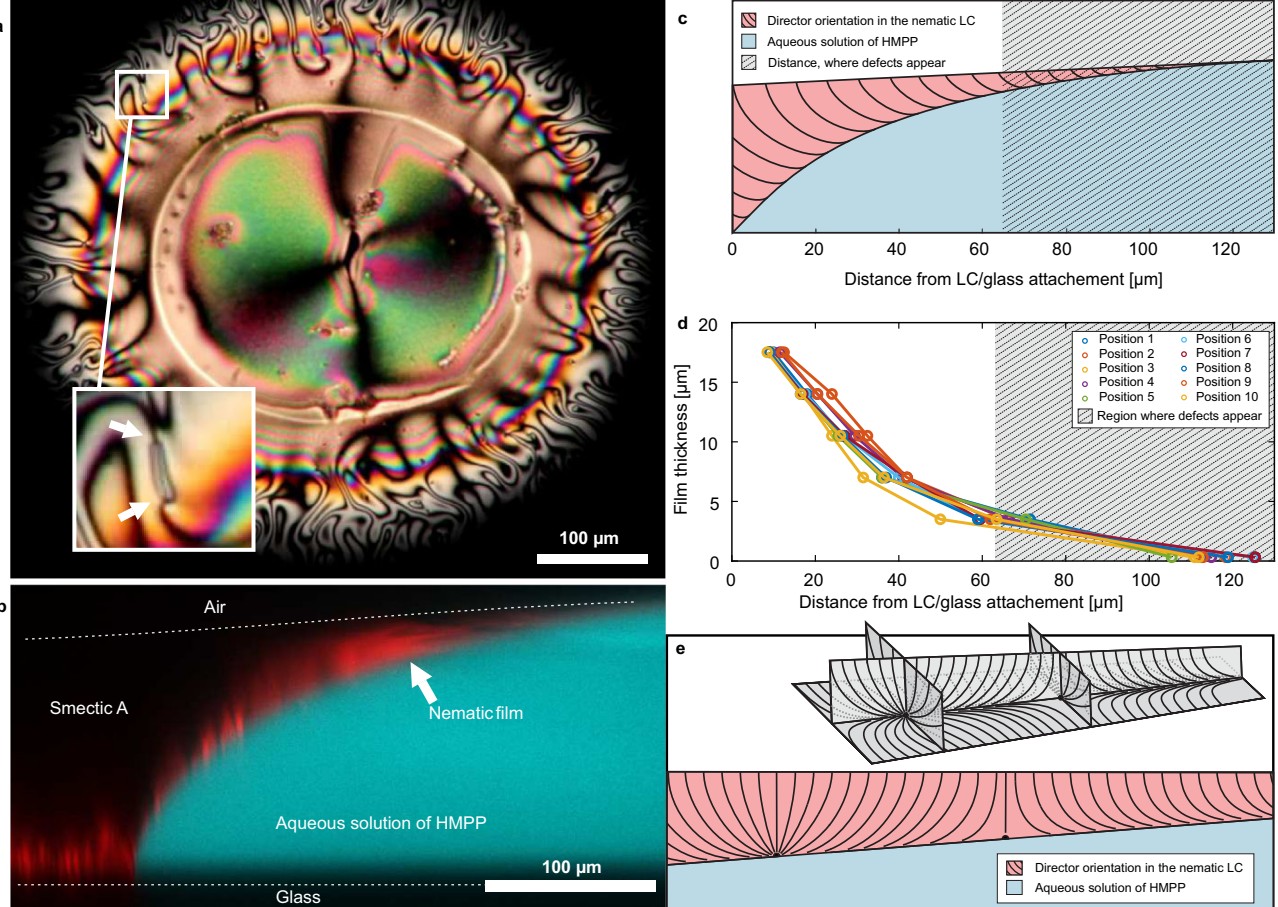

**Fig. 1 | Turbulence in a thin wedge-like film of liquid crystal (LC). a** A cross-polarized image of a LC film floating on aqueous solution of 2-hydroxy-2-methylpropiophenone photoinitiator (HMPP). Due to the continued diffusion of the HMPP from water into the LC, the film has turned nematic, although the temperature is 25 °C, where pure 8CB is smectic A. In the outer, corona-like and colorful band, topological defects can be observed, each as a set of four dark brushes emerging from a single point (marked in the inset). The colorful concentric fringes in the corona band suggest that it has the shape of a thin film that decreases in thickness when going radially outwards (see also Supplementary Movie 2).

**b** Vertical cross-section of the thin wedge-like nematic layer of 8CB, taken by fluorescent confocal microscope. Water, dyed with FITC, is shown in cyan (false color), while LC, dyed with Nile Red, is shown in red (also false color). The nematic film can be seen floating on and spreading along the surface of water. **c** Schematic representation of the LC film extending over the surface of the water droplet. The shaded area marks the part of the film where defects were formed. **d** Measurement of the thin film thickness at various positions around the drop shown in (**a**). **e** Schematic representation of the director in the film, containing a pair of defects, which are surface boojums.

---

Sooner or later they catch up with another defect of the opposite topological charge and annihilate along the way. Topological defects are on rare occasions also formed in the thicker hybrid nematic film that exists further away from the isotropic rim. An example of such an event is shown in Fig. 2(g–j) (for more examples see Supplementary Movie 3).

To identify the mechanism that is responsible for the creation of pairs of topological defects, we use the λ-plate polarizing microscopy technique that enables us to determine the local director orientation from the observed color of the nematic film. We modified the experimental geometry and cast the LC in shape of a line, as shown in the inset to Fig. 3(b). The water solution was cast to one side of the LC line. In this way we have straight boundaries throughout the observed area. Interestingly, there was no 'explosive' behavior in this case, as the LC was in contact with the water surface from the very beginning. This smooth evolution of the defect-formation process made the system more predictable and we were able to follow the nematic film's creation from the start, when the HMPP began entering the LC. We could see the nematic film growing and spreading along the water surface and becoming thicker on one side. Topological defects were not present from the start, but appeared after some time.

Fig. 3(a) shows a director configuration at the beginning of the experiment when the flow of the NLC is slow and laminar (see Supplementary Movie 4). We can clearly see the regions of the LC, colored yellow and blue, separated by thin red bands. In the top part of the image in Fig. 3(a) we can see a horizontal interface between the alternating yellow and blue regions and a disordered array of drop-like structures. Each of these spherical, drop-like objects of typically 10 μm diameter is a 'focal conic' domain, which is a characteristic defect for smectic A LC in a hybrid geometry[49]. The horizontal interface is therefore the nematic-smectic A interface.

The yellow and blue of the nematic LC regions in λ-plate images can be ascribed to two different orientations of LC molecules with respect to the axis of the polarizer, as schematically shown in the inset of Fig. 3(a). If the LC molecules are aligned along the axis of the polarizer, the color of this region will be red. However, if the molecules are rotated clockwise by an angle φ with respect to the axis of the polarizer, the region will look blue. And if the molecules are rotated anti-clockwise by an angle −φ with respect to the axis of the polarizer, the LC will appear yellow. This leads to the conclusion that the 1D-periodic texture in Fig. 3(a) is a herringbone-like periodic LC structure presented schematically in the inset of Fig. 3(a).

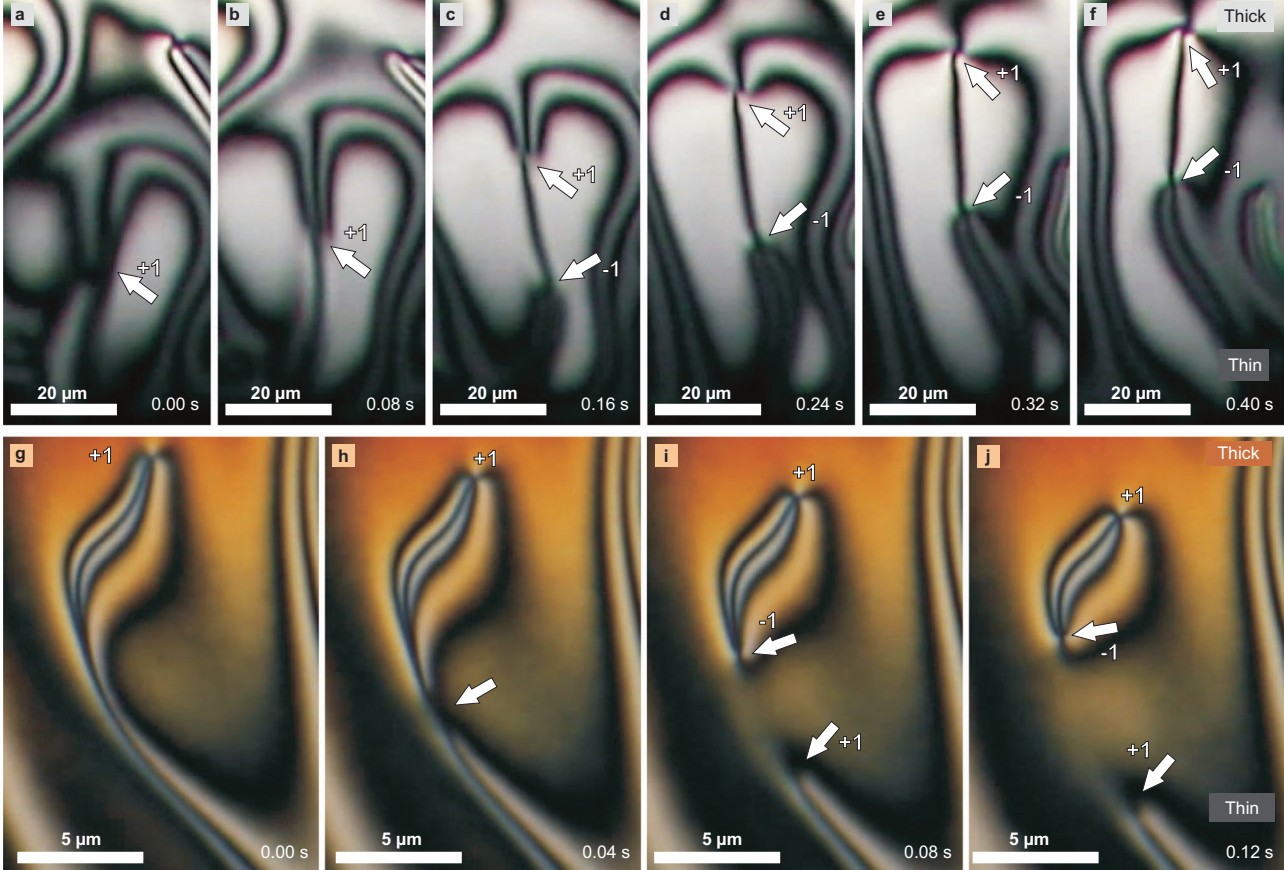

**Fig. 2 | Formation of topological defects with integer winding number.**
**a–f** Series of snapshots showing the formation of a pair of oppositely charged defects with an integer winding number. In a typical defect-formation event, an area spontaneously appears (see white arrow in (**a**)). As this area moves towards the core of the drop, it spontaneously splits into a pair of defects, which are clearly seen in panel (**c**). The two defects then travel upwards and usually annihilate, as the second one is faster and catches the first one. **g–j** Pairs of defects can also be created in the center of the nematic film, which is thicker. A pair of defects is spontaneously formed by 'cutting' the texture, see the pair of arrows, emerging from a single area indicated by the arrow in panel (**h**). The probability of defect-pair formation is higher in the thinner part of the film than in the thicker part of the film.

At a later time, additional 'islands' of red appear both in the yellow and blue domains, as shown by the yellow circle in Fig. 3(b). This red indicates that the angle $\varphi$ is close to 90° as measured from the direction of the polarizer, and the angle changes from the small value at the perimeter of the yellow (or blue) region to the nearly perpendicular orientation in the center of each region. As time increases, the LC film becomes thicker and new colors, corresponding to higher orders of retardation, can be seen (Fig. 3(c)). As the water evaporates with time, the HMPP water solution becomes saturated and starts forming droplets at the surface. These also enter the nematic film, causing the HMPP concentration to vary locally, which further adds to the chaotic motion inside the nematic film. As the behavior grows more chaotic, topological defects start to form as indicated in the last panel in Fig. 3(d). The system has gradually transited from the laminar flow at the beginning of the experiment in Fig. 3(a) to turbulent behavior in Fig. 3(d).

There is one very important observation to be made here about the various LC phases, which are seen in Fig. 3. There is a nematic phase on the bottom of each panel and the smectic A phase above the horizontal interface, and the temperature is the same throughout the sample and equal to 25 °C. As the appearance of the phase of 8CB depends on the local concentration of the HMPP, it is clear that the concentration of HMPP must be higher in the nematic phase (i.e., below the interface) and it must be lower in the smectic A phase. Clearly, there is a concentration gradient of HMPP molecules in the sample which must be the reason for the onset of counter-rotating flow

pattern, and at later times for the onset of turbulent flow, shown in Fig. 3(d).

The pattern of LC flow in the laminar regime, where the herringbone director pattern is observed, is revealed by putting small tracer particles (glass beads of 1.5 μm diameter) in the LC, and by following their movement using video capturing. We observed that the tracer particles always move only in the 'red' thin bands in between the blue and yellow regions, as schematically presented in Fig. 3(e). The tracer particle flows towards the thicker part of the LC via 'red channels', where a yellow region is on the left and a blue region is on the right, looking along the particle velocity. In the neighboring 'red channel' the particles move in the opposite direction, with the yellow region again on the left-hand side, and blue region on the right-hand side. It is then clear that the LC flow within the herringbone pattern has the form of 2D vortex rolls, as shown in Fig. 3(e). These vortex rolls are similar to Rayleigh-Bénard convection rolls, and are in our case driven by a concentration gradient.

The left-hand side of Fig. 3(e) shows part of the the yellow-blue NLC pattern, superposed with the schematic pattern of flow, which is strongest in the red channels between the yellow and blue domains. The right-hand side of Fig. 3 (e) shows the suggested director-field configuration in the yellow and blue domains, separated by red channels, where the flow is the strongest. This state is present in the initial stage of the experiment, where the flow is laminar and no topological defects are produced. Typically, this state develops a few minutes after the water with HMPP comes in contact with the LC. It is

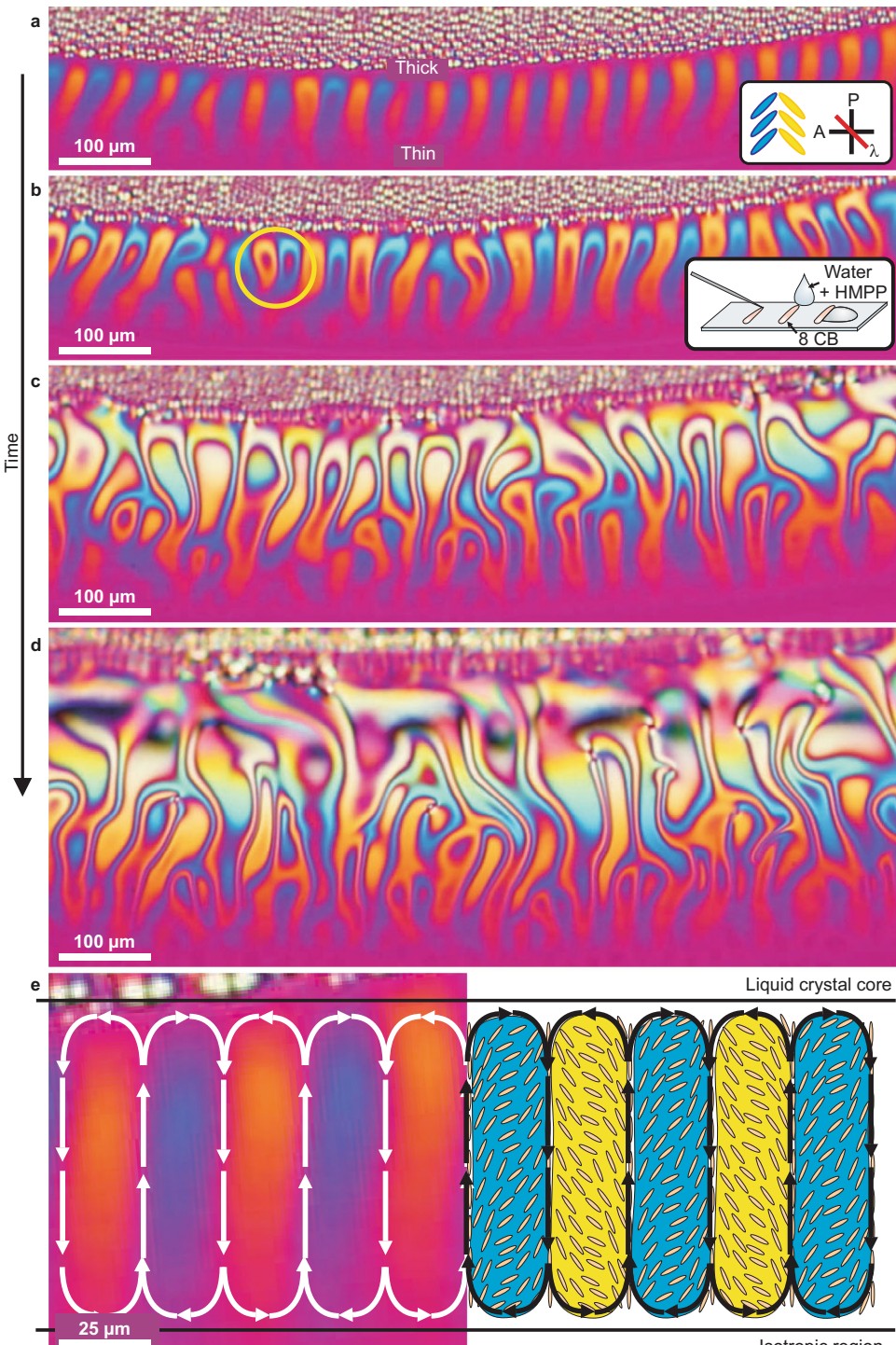

**Fig. 3 | Development of turbulence in a thin hybrid nematic layer. a** In the thin hybrid liquid crystal (LC) film the director gradually (after ~2 min) develops a periodic domain configuration, where blue and yellow regions correspond to areas where the nematic LC molecules are tilted in opposite directions with respect to the polarizers (as schematically shown in the inset). The blue and yellow regions hence represent two domains of the the herringbone-like pattern, shown in the inset. **b** Later, islands of opposite orientations are seen to emerge within the primary domains. The inset shows a changed experimental setting. HMPP - 2-hydroxy-2-methylpropiophenone photoinitiator. **c** The LC film thickens with time, resulting in a broader range of colors corresponding to higher orders of retardation. Islands within islands of the same color can be observed. The director field and flow become more chaotic. **d** The complexity of the director field increases further and topological defects start forming and annihilating. Panels (**a**–**d**) show the same area of the sample, the elapsed time between panels (**a**) and (**d**) is around 5 min. This time developement is also shown in Supplementary Movie 4. **e** Flow direction and schematic drawing of the director field at the initial stage of the experiment. The LC flows in a circular pattern, it moves towards the LC core at the junction where there is a yellow region on the left and a blue one on the right. At the opposite junction the LC flows outwards. On the right-hand part of the image, we depict the director corresponding to the colors in the λ-plate image.

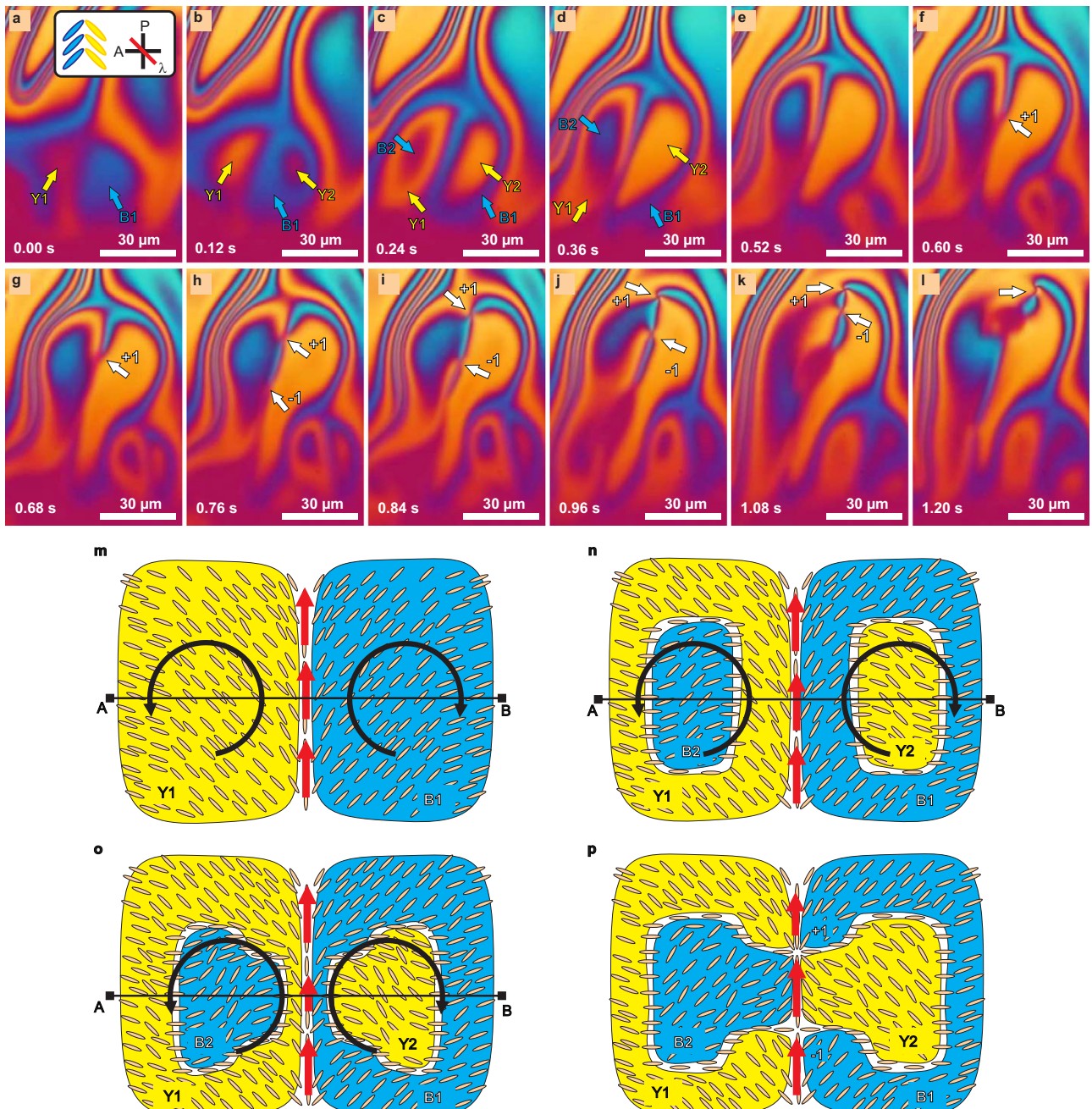

**Fig. 4 | Experimental and schematic display of topological defect creation.**
**a–l** The sequence of λ-plate images shows topological defect creation and annihilation. **a** Neighboring regions of yellow on the left (Y1) and blue on the right (B1) are formed. Between them the liquid crystal (LC) flows toward the LC core above. **b**, **c** Inside the primary regions, islands B2 and Y2 are formed. **d**, **e** When the flow velocity decreases, the neighboring strips of Y1 and B1 thin out. **f**, **g** The neighboring strips of Y1 and B1 disappear, which is followed by the creation of a defect. One defect (marked by a white arrow) can already be observed, while the other one only becomes observable in (**h**). **i–l** As they move, the defects come closer to each other until they finally annihilate. Panels (**a–l**) are snapshots from a video of Event 1 shown in Supplementary Movie 5. **m–p** Schematic view of defect formation shows the proposed director configuration while the defects are being created. **m** At the beginning the convection rolls are uniformly oriented. Red arrows symbolize the flow velocity along the line of maximum velocity, where defects will be formed. **n** Secondary regions with opposite orientations (B2 and Y2) are formed within the primary ones (Y1 and B1). **o** When the flow velocity is locally reduced, the neighboring bands of Y1 and B1 become thinner. **p** The thin stripes break and secondary regions B2 and Y2 become neighbors and topological defects are formed.

important to note that the orientation of the LC molecules in the thin red bands, where the flow is strongest, is parallel to the direction of liquid flow, and not perpendicular to it. This is also in line with the later observations when the topological defects start forming.

In Fig. 4 we show the time sequence of the defect formation when the overall director field was already very complex, with yellow and blue regions interchanging closely (see Supplementary Movie 5). In Fig. 4(a–c) we show how two neighboring areas – yellow (Y1) on the left

and blue (B1) on the right – appeared in the nematic, but at the isotropic edge. As the regions were moving towards the LC core, islands of different orientations and complementary colors (B2 and Y2) formed within the original two regions Y1 and B1. By comparing the molecular orientation in frames (a) and (b) in Fig. 4, we see that in the center of the blue region B1 the LC molecules have rotated clockwise and changed the color of the texture to yellow, whereas the LC molecules in the center of the neighboring yellow region Y1 have

rotated counter-clockwise and the texture changed color to blue. We have, therefore, two neighboring vortices centered in Y1 and B1 where the LC molecules collectively rotate in opposite directions. Because of this rotation, the center of each vortex winds-up and comes to the point where the neighboring strips of regions Y1 and B1 stretch and become thinner (Fig. 4(d, e)). Eventually, this red line breaks, allowing the secondary blue and yellow regions B2 and Y2 to come into contact (Fig. 4(f, g)). At this moment, a single and clearly visible topological defect is formed first, marked with a white arrow in Fig. 4(f). This first defect is a radial boojum with the winding number $k = +1$. Interestingly, the second defect, which is a hyperbolic boojum with the winding number $k = -1$, cannot yet be located at this moment, but becomes visible in a subsequent frame in Fig. 4(h). In Fig. 4(h–k) both topological defects move towards the LC core, while being attracted to each other. In Fig. 4(l) the defects are just about to annihilate, which they actually do at a later moment.

Figure 4(m–p) shows a simplified schematic view of the director during the process of defect formation, which we propose after carefully studying several defect-creation events, including the one in previous panels of Fig. 4. We can see following the molecular orientation across the AB line in Fig. 4(m) that the director is strongly splayed in the initial configuration. The molecules on each side of the red central channel rotate collectively in opposite directions, as indicated by the curved arrows in Fig. 4(m–o). The topological defects are created at the junction between the two convection rolls, where the LC flow is the strongest and directed toward the center of the LC. There the director is aligned with the flow, also pointing towards the LC's core at greater thickness of the hybrid aligned film. Red arrows in the image denote the flow velocity. In the initial stages neighboring regions of opposite orientations (Y1 and B1) are formed, as shown in Fig. 4(m). At a later time, islands of complementary orientation and complementary colors Y2 and B2 are formed within the primary regions Y1 and B1 because of the collective rotation of LC molecules. Figure 4(o) shows a local decrease in the flow velocity. This local decrease of flow velocity causes the neighboring bands of Y1 and B1 to stretch and become thinner.

The intermediate splayed stripe domain thins out to the point where it is cut and replaced by a stripe of opposite splay, as shown in Fig. 4(p). As a result, the inner islands B2 and Y2 merge into a single splayed domain with colors that are complementary to the original splayed domain in Fig. 4(m) and two topological defects are created. At first, a $+1$ topological defect is formed, which is followed by the emergence of a $-1$ defect shortly after, as observed in the experiments.

## Motion of topological defects

To monitor and measure the flow of the LC, we added 1.5-µm-diameter silica particles to the LC, which served as visible tracers, moving with the flow of the LC. The tracer particles must be small not to perturb the ordering of the LC. On the other hand, they have to be visible under an optical microscope. The density of the tracer particles is very small, which means they are separated from each other by tens of micrometers and therefore do not interact elastically with each other. Because of their smallness, the tracer particles are also weakly coupled to the local elastic deformation of the LC, which means that the main force that causes their movement is the drag viscous force of the flowing LC. By tracing the trajectories of these tracer particles we can map the flow velocity of the LC.

We selected a few typical videos of defect formation in a turbulent flow and manually tracked the defects and tracers. We found that in between different experiments, the flow velocities could differ by an order of magnitude as the HMPP concentration gradient driving the flow was time dependent and also strongly depended on the size of the system. In a narrow time window within the same experiment, however, the typical absolute velocities of the defects and tracers were similar in magnitude, though the absolute velocities of the defects had

a much wider distribution. We assume that this was because the microparticles were less affected by elasticity and more strictly followed the flow, driven by the concentration gradients. When a defect came close to another defect, it sped-up or slowed down relative to the underlying flow as it experienced the attractive elastic force of the neighboring topological defect. Since defects usually annihilated quickly, a large part of their movement consisted of approaching another defect and was thus governed by the elastic force of pair attraction. Distributions of the velocities of defects and particles are shown in Fig. 5(b, c) for a typical 25 s video, together with a frame of the video in Fig. 5(a). We also measured the angular distributions of defects' and tracers' velocities, which are shown in the insets.

As can be seen from the insets of Fig. 5(b, c), the defects' movements show narrower angular distribution than those of the tracer microparticles. The defects tend to move towards the LC core (this direction is indicated by 90°); almost none of the defects move in the opposite direction. This can be explained by the fact that the defects were mostly formed in a thin LC layer at the isotropic-nematic border of the LC film, and they can only move towards the thicker part of the LC layer. While traveling toward the LC core the defect pairs annihilate quickly, and no defects moving in the opposite direction could be observed. In contrast, the tracer particles could be tracked for a long time. They moved equally likely to and from the LC core (directions denoted by 90° and 270°, respectively). In both angular distributions we can see a preference for moving to the left, suggesting some additional flow in that direction.

By comparing the velocity distribution for the defects in Fig. 5(b) to the velocity distribution for the tracer particles in Fig. 5(c) we can safely conclude that the defects do not move solely due to the diffusion-guided material flow, as their flow velocities are different from those of the tracer particles. This means that their mutual elastic interaction gives rise to substantial dynamics of the defects. We also analyzed velocity distributions of + and − defects separately, which is discussed in Supplementary Note 2. The negative defects move apparently faster than the positive ones during annihilation, which is not in agreement with many experimental reports, where positive defects always moved faster than the negative ones. This can be explained by the overall drift of both positive and negative defects due to the background flow of the LC, which actually carries both defects.

## Numerical simulations

The mechanisms of continuous topological defect creation and annihilation are explored by using numerical modeling and combined theoretical analysis. We use a minimal model of the observed topological defect dynamics – regular and chaotic – as governed by the dynamic interplay of three main material fields: (i) nematic orientational order, (ii) photoinitiator concentration, and (iii) material flow. Each of these fields is determined by its own dynamic equation, but which importantly depends on the dynamics of the other two fields. The photoinitiator concentration field is modeled by the convection-diffusion equation, where the dynamics in the concentration is centrally affected by the advection of the concentration field with the material flow. The dynamics of the nematic orientational order is described within an effective two-dimensional nematic model that corresponds to the experimentally observed defect generating layer. The nematic alignment is described by the orientation vector field $\mathbf{n}$ (which note is a full vector due to hybrid nematic alignment out of the plane[50]) and the order parameter $S$ that measures the degree of molecular alignment. The used dynamics of the nematic alignment in the simulation locally corresponds to tensorial description of nematodynamics; however, the vectorial director field that we use in our calculations allows for the emergence of only *integer* topological defects. The assumed incompressible material flow is described by the Navier-Stokes equation with included forcing from the concentration field. The mutual strength of coupling between the three main material

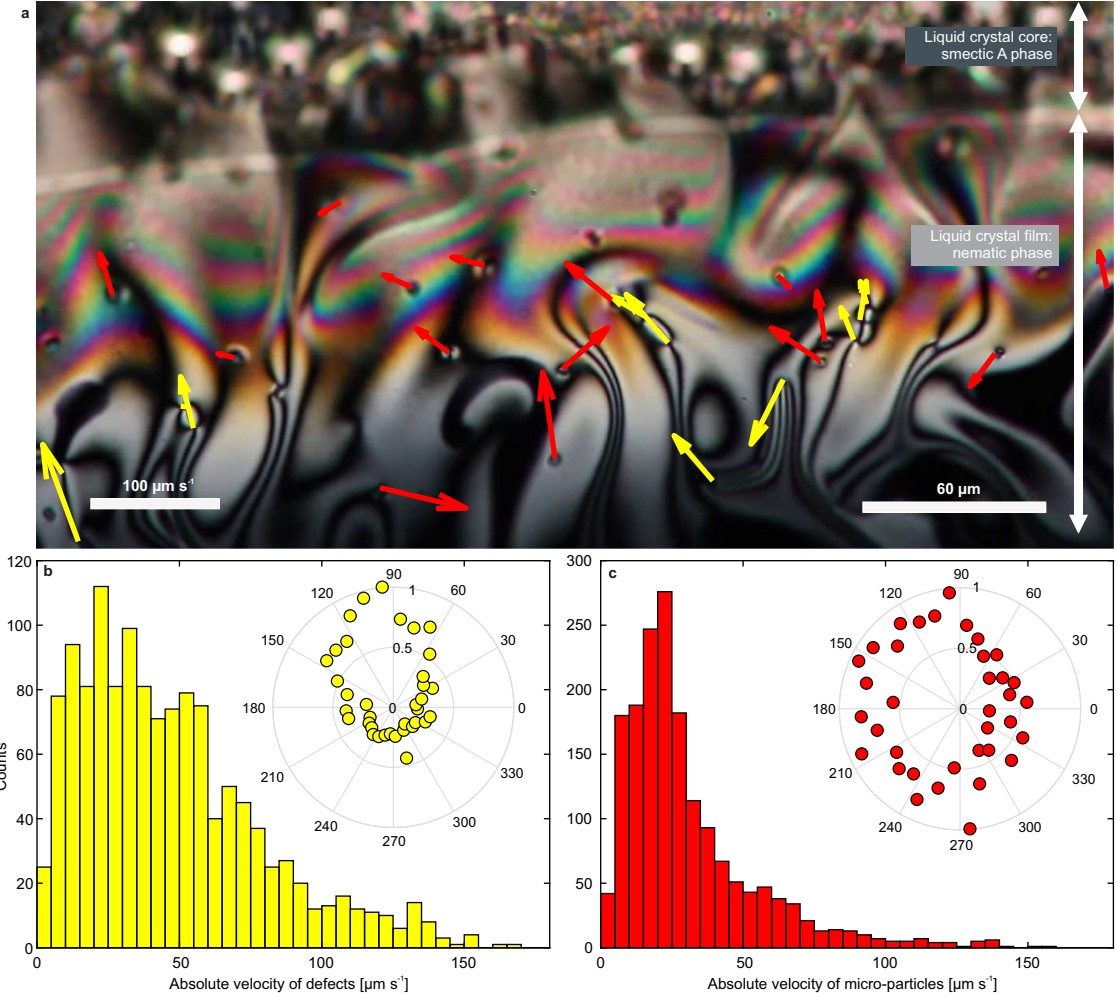

**Fig. 5 | Velocity analysis of defect and tracer movement in a typical 25 s video.**
**a** A frame from the analyzed video. Yellow arrows present the tracked defects' velocities and red arrows the tracked velocities of tracer particles. **b** Distribution of absolute velocities of defects. Angular distribution of measured defect movements is shown in a polar plot in the inset. **c** Distribution of absolute velocities of tracer particles. The polar plot shows the angular distribution of measured tracer movements.

fields can be characterized by two dimensionless numbers, the Rayleigh number (Ra) which measures the relative strenght of advection vs. diffusion, and the Ericksen number (Er) which gives the relative strength of material flow vs. nematic ordering. More specifically, the complete set of dynamic equations and how they are solved is outlined and explained in the Supplementary Note 3.

Using numerical modeling, we observe the emergence of two key limiting dynamic regimes, actually in good qualitative alignment with the experiments. At low Rayleigh numbers (which corresponds to weak flow driving, such as at early times of the experiments), a stable array of material flow vortex rolls forms as shown in Fig. 6 and in Supplementary Movie 6. The neighboring vortices are counter-rotating and are propelled by the photoinititator concentration field gradients (Fig. 6(a)). The emergent material flow vortices in-turn induce tumbling elastic deformation of the nematic orientational field, which is penalized by the nematic elasticity, which creates a dynamic steady state herringbone structure, shown in Fig. 6(b), and observed experimentally. At this low Rayleigh number regime there are no topological defects emergent within the nematic orientational field and the degree of order is approximately constant, as shown in the colormap in Fig. 6(b). Supplementary Movie 7 shows larger distortions of the nematic alignment at increased flow magnitudes, but still no topological defects are observed. Differently, at high Rayleigh numbers (i.e. high flow driving, such as at later times in the experiments), the

material vortex flow becomes much stronger in magnitude, creating large shear gradients, but more importantly it also becomes irregular. The flow vortices emerge and move irregularly in the layer, as directly coupled with the irregular patterns of the photoinitiator concentration field (see Fig. 7 and Supplementary Movie 8.). This irregular – chaotic – dynamics in the flow field leads in turn to the irregular nematic orientational dynamics, that results in the formation of the topological defects, as shown in Fig. 7(b). As observed experimentally, we see emergence of integer +1 and −1 winding number defects, which exhibit chaotic turbulent dynamics. Indeed, again as observed in experiments, the topological defects are created in opposite charged pairs, which assures the conservation of the net zero of the topological charge. Figure 7 shows the generation of the topological defects in time, where an increase in the local driving (concentration of the photoinitiator) creates a larger material flow, which in turn deforms the nematic orientational field until a defect pair is produced, to relieve some of the nematic elastic distortion. Figure 8 shows in more detailes simulation of a process, where pair of topological defects is formed, in remarkable agreement with experiments. Importantly, this demonstrated instability for the defect pair generation is seen only in the tumbling regime of the nematic, directly underlying the importance of the material flow to nematic ordering coupling.

The Ericksen number as the second main material coupling measure is observed to affect the degree of nematic deformation in the

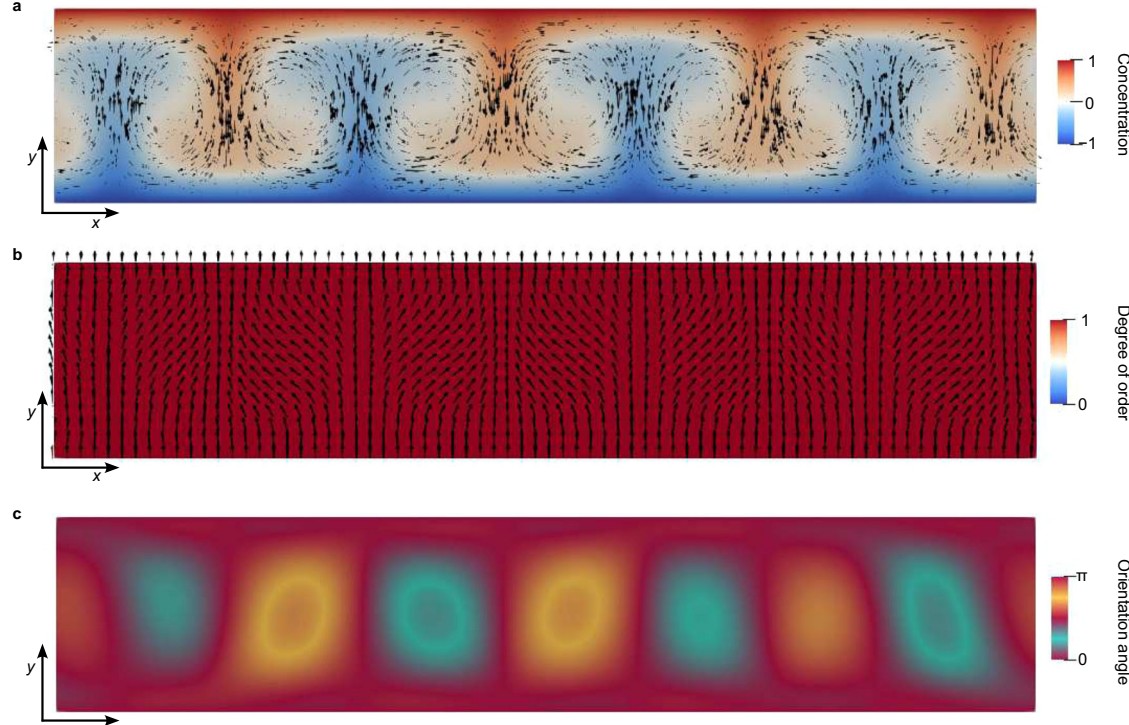

**Fig. 6 | Stable periodic vortex-roll structure at Ra ≈ 60 and Er ≈ 12.**
**a** Concentration field (colormap) and velocity field (black arrows) form periodic vortex flows. **b** The orientational field (black arrows) is periodically distorted in the vortex flow. **c** Colormap of the absolute value of the orientational angle |α|, where

$\mathbf{n} = (\cos \alpha, \sin \alpha)$ and $\alpha \in [0, \pi]$ matches the color palette of the red-plate micro-scopy in Fig. 4. Periodic distortions of yellow and blue correspond to the periodic deformation of the orientational field. This figure is a snapshot from Supplementary Movie 6.

flow field. At low Ericksen number (Er ≈ 12 in Fig. 6), the tumbling deformation of the orientational field is counteracted by the nematic elasticity and no defects are observed in the herringbone structure. At higher Ericksen number (Er ≈ 130 in Fig. 7 and Fig. 8), the velocity field is able to deform the orientational structure to produce pairs of integer-charged defects.

## Discussion

This is one of the rare realizations of a passively driven soft-matter system where topological defects are continuously produced and annihilated, realizing a topological dynamic steady state, in strong difference to the equilibrium topological states and structures which are better known. In some aspects it resembles 2D active nematics[32], where topological defects are continuously produced and annihilated by consuming chemical energy with molecular motors that drive continuously the extension of micro-tubulin fibers. In our system, defect creation is driven by the vortex-like flow of the LC, which is driven by the dynamics of the foreign molecules in the LC. The difference in the driving mechanism is also responsible for the different types of topological defects that are created in active nematics and nematics driven by the concentration gradient. We compare the two mechanisms of the topological defect's creation in Fig. 9(a, b).

Fig. 9(a) illustrates the mechanism of topological defects' creation in active nematics based on the bend instability that was observed in experimental systems such as a dense water solution of microtubules[32]. In this case the nematic phase is obtained by sponta-neous ordering of the microtubules, which are interconnected with kinesin molecular clusters that act as molecular motors, pushing the two neighboring microtubules in opposite directions. With increasing molecular motor activity, this extensile motion eventually leads to the self-amplifying of bend distortion (buckling instability), as illustrated

in Fig. 9(a). The director locally bends until it breaks at some moment and this results in the creation of a pair of −1/2 and +1/2 defects. The region where the director field breaks is highlighted yellow in Fig. 9(a).

In our case the scenario for defect production is different and is illustrated in Fig. 9(b). Due to concentration gradient of HMPP pho-toinitiator, a system of counter-rotating flow vortex rolls is developed. This flow profile – in the tumbling effectively polar nematic ordering – locally gives rise to the onset of splay deformation of the director pattern, shown in the second panel in Fig. 9(b). Over time this splay deformation is amplified to the point where the polar director-field lines break, as illustrated in the 4th panel in Fig. 9(b). Close observation of the flow reveals that at that moment the local velocity of the LC flow is slowed down, and is followed by the creation of a pair of topological defects with +1 and −1 winding, as shown in the last panel of Fig. 9(b). Defects are therefore formed due to a local decrease of the absolute velocity of the flow, which effectively stretches the director-field lines and a pair of defects appears, one shortly after the other. In all the experiments only defects, characterized by four dark brushes, were seen, and four brushes can be ascribed only to |1| winding number defects.

Whereas this mechanical picture explains the formation of topo-logical defects, the physics that is behind this process is not obvious. The flow of the LC must be driven by a force, or several forces that are related to the local concentration of the photoinitiator. They can be divided into two classes: (i) surface forces akin to Marangoni force, generated by a change in surface tension due to presence of surfac-tants, and (ii) bulk forces generated by the diffusion of small molecules through the volume of the LC, i.e. the bulk force due to osmotic flow. These forces are the generators of fluid flow.

The effective surface force can be written as a sum of the fol-lowing contributions, which notably are conditioned by the fact that the actual dynamics in experiments occurs over a wedge volume

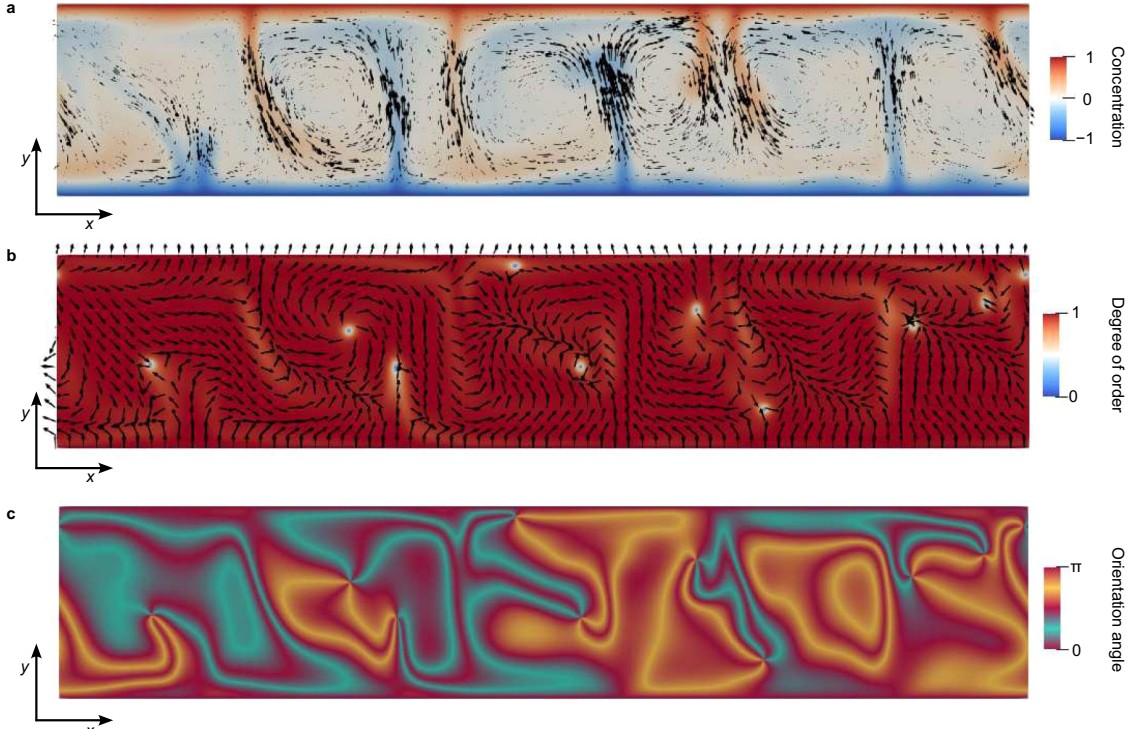

**Fig. 7 | Dynamic nematic regime at strong flow driving (high Ra number).**
**a** Chaotic flow field (black arrows) emerges with irregular structures of flow vortices. The colormap shows the corresponding concentration field of the photoinitiator. **b** Topological defects emerge in the nematic orientational field, as seen both in the vectorial orientational nematic field (black arrows) and in the nematic degree of order (colormap). Note the total $5 + 5 = 10$ *even* number of defects (5 of topological charge −1 and 5 of topological charge +1). **c** Corresponding optical image of the nematic orientational order in (**b**). Calculations are performed for Ra ≈ 650 and Er ≈ 130. Panels are snapshots for set time from Supplementary Movie 8.

(and not a flat layer; see also Supplementary Note 3 and Supplementary Fig. 4):

$$\mathbf{f} = k\nabla\phi + \frac{\sigma_0}{h}\nabla h + \frac{k}{h}\phi\nabla h, \tag{1}$$

where $h$ is the local thickness of the nematic film and $k = \frac{\partial\sigma}{\partial\phi}$ describes the dependence of the surface tension $\sigma$ on the concentration $\phi$ of the HMPP photoinitiator. The first term in Eq. (1) is the Marangoni force, which acts on each interface and it pulls the liquid towards regions with larger surface tension $\sigma$. Besides the Marangoni force, there are two terms in force density in Eq. (1) that depend on local nematic layer thickness $h$. $\frac{\sigma_0}{h}\nabla h$ is the surface force due to wedge-shape (geometry) of liquid (nematic) layer that drives the liquid towards regions of higher surface tension $\sigma$. The third surface force term combines Marangoni and geometry-dependent terms.

We performed extensive numerical simulations by varying the sign and magnitude of individual surface force terms in Eq. (1) and we observe that Marangoni force can induce irregular/chaotic flow and defect production only if $k = \frac{\partial\sigma}{\partial\phi} > 0$. This means that the surface (or interfacial) tension must increase with increasing concentration $\phi$ of photoinitiator. As described in Supplementary Note 4, we measured surface tensions of LC with HMPP (Supplementary Fig. 8), water with HMPP (Supplementary Fig. 9) and in all cases surface tension decreases with increasing concentration which is a common effect of surfactants. Within the limitations of the presented minimal model and for the scanned parameter range (though quite extensive), this comparison between experiments and theory implies that Marangoni force is not likely to induce topological turbulence in our experiments.

The second possibly relevant mechanism for the emergent material flow is the diffusion and mixing of photoinitiator within the bulk of the liquid crystal layer, which can be described by volume force

terms that correspond to the free energy density of mixing[51]. The corresponding volume density force is

$$\mathbf{f} = -\phi\nabla\mu, \tag{2}$$

where $\mu = \delta f/\delta\phi$ is the chemical potential of the mixture. For different models of the chemical potential on the concentration density, different bulk force terms can be calculated from Eq. (2). Among them, we performed calculations and observed that the terms $\frac{1}{h}\phi(\nabla\phi)$, $-\frac{1}{h}\phi(\nabla\phi)$, $\phi\nabla(\nabla^2\phi)$ are able to induce vortex material flow within our minimal model, whereas terms $\phi(\nabla\phi)$, $-\phi(\nabla^2\phi)$ were not. To possibly discriminate the relevance of these mixing terms, extensive studies and also measurements of the chemical potential in our system would be needed as depended on the concentration field, likely at first stages performed in simplified geometries and experimental setups. Overall, we identify surface Marangoni forces and bulk forces due to surfactant mixing as main candidates for the driving mechanisms of the defect production. Within the presented minimal model, the surface forces are not seen as capable of generating the instability to the irregular flow of the topological turbulence as observed in our experiments. Most likely, the turbulence is triggered by a combination of several surface or bulk forces due to the diffusion and mixing dynamics in the concentration of the photoinitiator. We can also exclude the generation of radicals due to illumination of photoinitiator with light. We observed that radicals can be generated by UV illumination, but not with white light of microscope lamp. The experiments performed with white or red illuminating light also showed no difference. The turbulence could not be achieved using other surfactants, such as oleic acid, acetone, sodium dodecyl sulfate (SDS) or soap. Finally, we observed production of topological defects due to HMPP concentration gradient also in 7CB liquid crystal, but not in 5CB. This suggests

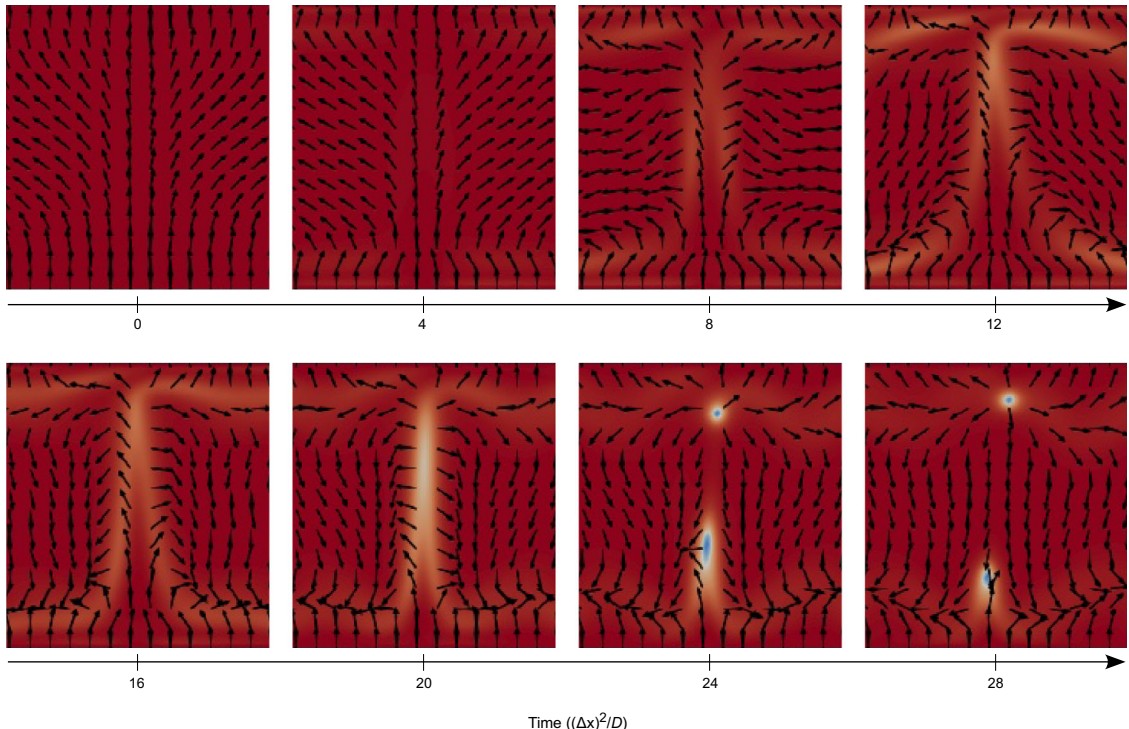

Time $((\Delta x)^2/D)$

**Fig. 8 | Generation of topological defect +1 and −1 pair in time as caused by the irregular-chaotic flow at high Ra number.** Increased local driving (concentration of photoinitiator) generates stronger material flow which in turn deforms the nematic orientational order until a defect pair is produced, relieving some of the elastic distortion. Locally, flow is oriented along the +$y$ direction and the +1 defect is generated at the top and a −1 defect at the bottom. Full dynamics is shown in Supplementary Movie 8.

that the mechanism is strongly related to molecular structure: both 7CB and 8CB are tumbling nematics, while 5CB is flow-aligning nematic.

In conclusion, the examined, passively driven, hybrid nematic LC film exhibits continuous spontaneous defect formation and annihilation over a long timescale. We presented only surface point defects, but it could also serve for studies of spontaneous creation, movement, rewiring, and annihilation of defect loops, which were sometimes detected in the thicker part of the LC film, emanating from the smectic LC core. We believe that the same mechanism of topological charge production could be realized in chiral nematics, where we expect that much richer topological phenomena would be observed. Finally, at this stage, the presented results are a possible indication of unknown exciting research into controlling and realising structures of chemical potential, as enabled by the structure and topology of the complex fluids.

## Methods
### Materials and sample preparation
We used a common smectic LC component, 8CB (4-octyl-4′-cyano-biphenyl, by Frinton Laboratories, Inc.). At room temperature it exhibits the smectic A phase and has a phase transition to the nematic phase at 33 °C and to the isotropic phase at 40 °C. We cast the LC onto a glass slide with a needle, either in the shape of a circular drop or in the shape of a line. The LC was then put into contact with ultra-pure water containing 0.4 wt% of 2-hydroxy-2-methylpropiophenone photoinitiator (HMPP, Irgacure 1173 by Sigma Aldrich Co.). The aqueous solution was either cast on top of the LC drop or beside the LC line. The HMPP is a substance that is poorly soluble in water, with 1 wt% being close to a saturated solution. It is more soluble in the LC. That is why the HMPP starts to diffuse from the water medium into the LC, which causes a decrease in the LC's order.

For the characterization of the flow we added 1.5-µm silica microspheres (from Bangs Laboratories, Inc.) into the LC as tracers. The microparticles were treated with DMOAP (Dimethyloctadecyl[3-(trimethoxysilyl)propyl]ammonium chloride, by ABCR GmbH) prior to use, to achieve homeotropic anchoring.

### Film-thickness estimation
The film thickness was estimated from the interference colors in the cross-polarized images. At various radial cross-sections around a drop of LC we determined the distance of magenta-colored parts from the position of the LC attachment to the glass. These corresponded to subsequent orders of retardation (550 nm, 1100 nm, 1650 nm, etc.). Additionally, we determined the distance at which we could observe the transition from black and white to yellowish, which corresponded to a retardation of 50 nm. The film thickness can be roughly estimated as an optical retardation divided by the material birefringence. What we get is a low-end estimate for the thickness as only the in-plane component of the director contributes to the optical retardation. Since we have hybrid anchoring conditions, the upper part of the film with the vertically aligned molecules does not add to the thickness estimate, although in reality it probably makes a substantial contribution to the overall film thickness.

### Microscopy
The sample was observed with a polarization microscope (Nikon Eclipse E600 POL). Cross-polarized and $\lambda$-plate images and videos were taken with a CCD camera (Canon EOS 550D). To see a vertical cross-section through the sample we used a confocal microscope (Leica TCS SP5 X). For confocal imaging we dyed the 8CB liquid crystal with Nile red dye (by Sigma Aldrich Co., emission maximum 603 nm (in 8CB)) and the aqueous solution of HMPP was dyed with Fluorescein-5-isothiocyate (FITC, by Invitrogen, emission maximum 519 nm). These two dyes are spectrally separated

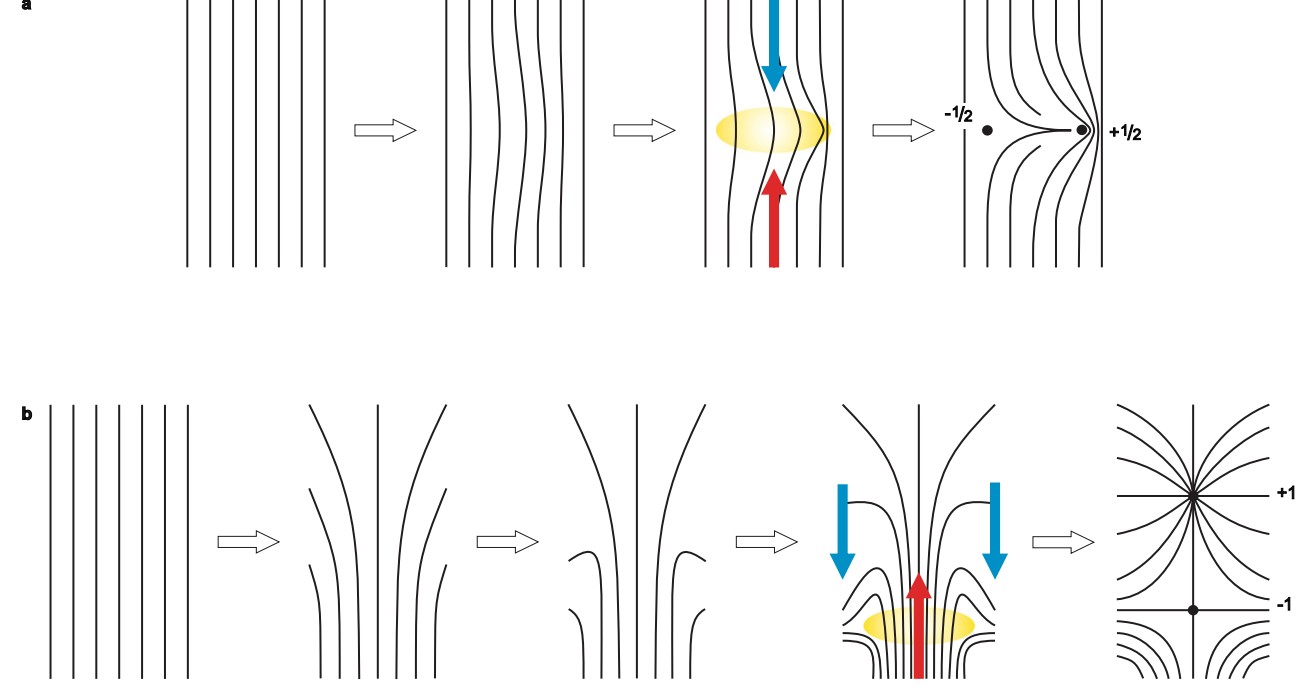

**Fig. 9 | Production of topological defects in 2D active nematic and in our layer of driven nematic liquid crystal (LC). a** In the active nematic LC a pair of −1/2 and +1/2 topological defects is created by the self-amplifying bend instability. The instability emerges from a bend fluctuation in the yellow area, which is amplified by the activity and at some instant gets relieved by the formation of the defect pair of opposite half-integer winding. **b** In driven nematic LC, the counter-rotating flow vortices give rise to self-amplifying splay distortion. The counter propagating flow indicated by the blue and red arrows gives rise to local shear flow that frustrates and ultimately breaks the polar vectorial orientation field in the yellow-shaded region. This gives rise to a pair of +1 and −1 winding number topological defects, with +1 defect drifting with the LC flow in front of −1 defect. Therefore, the difference between (**a**) and (**b**) is in the symmetry of the flow pattern, in the type of the elastic mode (bend vs. splay) and in the symmetry of the order parameter (nematic vs polar).

enough to be imaged independently. In images the two independent signals are superimposed and artificially colored (LC - red, water - cyan).

### Surface tension measurements

Surface tensions were measured by applying a pressure to a glass capillary containing the liquid and inflating a liquid droplet into water or air. The capillary was attached to the microfluidic flow controller (Elveflow, OB1, 0-200 mbar), with which pressures of a few millibar could be applied with a 0.03 mbar precision. For each value of the applied pressure we measured the radius $R_1$ of the inflated droplet and the radius $R_2$ of the interface between the liquid (LC or water) and compressed air in the inner part of the capillary, which is used to inflate the droplet. For each liquid (8CB or water), surface tension was calculated from the balance of forces due to Laplace pressures at two curved interfaces and the applied pressure.

### Data availability

The experimental source data and data generated in this study have been deposited in the Zenodo database under accession code https://doi.org/10.5281/zenodo.7193782. All numerical simulations data generated in this study are provided in the paper, Supplementary Information and Supplementary Movies.

### Code availability

The code developed in this study for analysis of experimental data has been deposited in the Zenodo database under accession code https://doi.org/10.5281/zenodo.7193782. The numerical methodology code used in this study is available in Zenodo database under accession code https://doi.org/10.5281/zenodo.4737814, with additional details available at reasonable request.

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

## Acknowledgements

The authors would like to thank Julia Yeomans, Amin Doostmohamadi, and Simon Čopar for fruitful discussions and useful suggestions at various stages of this work. The authors acknowledge financial support of Slovenian Research Agency (ARRS) through grants: PR-05543 (M.M.), P1-0099 (Ž.K., M.R. and I.M.), N1-0124 (Ž.K.), J1-1697 (I.M., M.R.), J1-2462 (M.R., I.M.), and N1-0195 (M.R.). I.M. and M.R. acknowledge support from the European Research Council under the European Union's Horizon 2020 research and innovation program (Grant Agreement No. 884928-LOGOS).

## Author contributions

M.M. preformed experiments, analyzed the results and wrote the first version of the manuscript. Ž.K. performed numerical calculations. I.M. supervised the experiments and revised the manuscript. M.R. supervised numerical simulations and wrote the numerical part of the manuscript with Ž.K. All authors contributed to the final version of the manuscript.

## Competing interests

The authors declare no competing interests.
