## [Peer Review File · Nature Communications]

Continuous generation of topological defects in a passively driven nematic liquid crystalREVIEWER COMMENTS

Reviewer #1 (Remarks to the Author):

The present manuscript investigates in detail the continuous formation and annihilation of topological defects in a thin liquid crystal film in contact with a fluid during driven turbulent flow. The paper is very well written, concise and informative, and should be of interest to a wide soft matter community, also due its relation to active materials. The well-referenced introduction provides the explanations needed by non-specialists reading the paper. Detailed descriptions are provided of the experiment so that it could be reproduced easily by other groups. The discussion and interpretation of the presented results are reasonable and verified by the data and movies provided. Given the lately broad interest in topology, active materials and defect creation and annihilation, I believe that this manuscript would make a good contribution to Nature Comm. and I thus would recommend publication.

I would just like to address a few issues that the authors may want to consider:

1. When reading the paper up to fig.2 the question arises which one of the defects is the +1 and which is the -1 defect. I do realise that this is explained later, but it might help the reader if this is already indicated in fig.2.
2. During the annihilation process of the defect pair, is there an anisotropy in the defect dynamics, as it is seen in other experiments? And if so, is this comparable? A short discussion of this aspect might enrich the paper somewhat.
3. Also in fig 6(d) one could indicate which one of the defects is +1 and which is -1, because readers outside of the field may not be instantly familiar with the director configuration of both defects.

Reviewer #2 (Remarks to the Author):

The manuscript "Continuous generation of topological defects in a passively driven nematic liquid crystal" by Mur and Musevic depicts topological defect generation in dynamic flow in nematic liquid crystal confined in a hybrid anchoring condition. There are mainly two parts describing different experiments, i. e. 1) The dynamic structural transformation (explosion) of a hemispherical droplet of hybrid SmA/N state, and 2) defect formation in nematic thin-films upon passive dynamic flow (convection) driven by diffusion of small molecules. These two parts are linked, because the dynamics in the latter part was found by the observation of the former process. But, as far as described in the manuscript, they are not very much correlated. As a result, the main point in the present manuscript became unclear, and thus in this context, the storyline needs improvement. As given in the title, the main topic of the present manuscript is the latter, i.e. the generation of the topological defects via the passive flow. However, this part is still too qualitative. With the aid of video analysis, the authors succeeded in describing the generation process of defects, together with the flow field distribution. Intuitively, the conclusion simply says, the interplay between the elasticity and the diffusion of the small PI molecules transferred from water phase to the N phase is the initial driving force of the flow, and non-equilibrium dynamics due to the flow-elasticity coupling is a role to generate such a unique defect formation state. But it looks almost all in the present manuscript. Still it would be difficult for the readers to catch what was the newly-understood physics or more broadly, science brought by these experimental data and analyses. To describe more the physics, the authors may need more quantitative physical analysis as they admitted by mentioning "This means that their mutual elastic interaction gives rise to substantial dynamics of the defects. However, to say more about the elastic

contribution to the flow dynamics of defects we need to know the flow velocity field of the LC in real time". So, while I like the concept of the present work as well as high quality data, the manuscript in this stage is still too weak because of lack of new physics, and it is difficult to publish the present manuscript in Nature Communication.

Apart from the above, I have some specific/technical questions;

1. In this manuscript, still the relation between the surface condition and the flow dynamics are not clear. In addition, I wonder if the Marangoni effect at the interface is also playing for the process, or fully bulk diffusing flow? If so, is it possible to be further discussed with some kind of physical analysis, like surface tension measurement? But also it would be nice to mention more the diffusion and the internal flow.

2. Related to 1, is there any method to make it possible to visualize the distribution of the PI concentration and surface tension?

3. In the present case, is the use of 8OCB having the N-SmA transition necessary for the induction of this defect generation phenomenon? I thought that the SmA core in the droplet is important for the explosion process, but it is not clear how it works for thin-film expansion and defect formation, especially in the case of the stripe experiment. Or can we use any Nematics for the same experiment?

Reviewer #3 (Remarks to the Author):

This article titled "Continuous generation of topological defects in a passively driven nematic liquid crystal" investigated the continuous generation of topological defects with an integer topological charge. The approach of generating topological defects in liquid crystals (LCs) by the gradual diffusion of foreign substances at the water-LC interface is considered a counterpart for active nematic systems using biomaterials. The mechanisms and experimental methods for generating turbulence are new, and the qualitative analysis of the formation and behavior of topological defects is well discussed. This study is expected to appeal to a broad readership of 'Nature Communications'. However, some parts need to be supplemented, and if the authors can discuss the contents of the following questions in the research results, the reviewer agrees to be published after the revision.

*Major questions

1) Authors utilize "2-hydroxy-2-methylpropiophenone photoinitiator (PI)" as foreign material for generating turbulence. However, the author did not explain why this specific foreign material was used. Is there any reason to select it? Do other substances that can dissolve in water and diffuse into LCs show the same experimental results?

2) From a chemical point of view, photoinitiators generate radicals when exposed to lights as a follow-up question. It is thought that this could also happen during optical microscopic observation, wouldn't this be an additional source of turbulence or 'explosion'? Please describe the presence or absence of radical formation or its effect.

3) The authors stated that the flow of a fluid changes over time from laminar to turbulent. Is it possible to quantitatively explain (e.g., Reynolds number) when this transition occurs using the characteristics and thickness of the LCs?

4) The temperature of the experiment affects both the diffusion rate of the photoinitiator and

the phase transition temperature of LCs. It means that changing the experimental temperature can change the velocity and frequency of appearance of topological defects. Please verify this in the revised manuscript. And it might have the same function as controlling the velocity of bacteria in the active nematics systems; please discuss this.

5) The authors analyzed the velocity and direction of turbulence in the LC by adding fluorescent particles. It might help readers understand if authors add vector fields with a color map.

6) The author explained the reason for using 8CB in the introduction, including references. If 5CB is used under the same experimental conditions, are not topological defects formed at the isotropic and nematic interface? If not, how does it behave?

***Minor questions**

1) This research can be classified as an LC research field. Thus, the abbreviation photoinitiator (PI) reminds many readers of polyimide. For this reason, it is recommended a different abbreviation.

2) The author provided high-quality videos of topological defects. However, defects formed in the videos cannot be easily observed. So, it is recommended to add some markers in the video for topological defects.

3) The article is well organized, but the same concept seems to be explained with too many duplicates in the manuscript.

Response Letter to Reviewers' comments and criticism

We thank the reviewers for careful reading of the article, constructive criticism and positive comments. We agree with all the criticism raised by the referees. We have followed their recommendations and made substantial revision of our original work. We also added an extensive Supplementary Information with 3 movies showing simulation results. All changes in the revised manuscript are in red text.

We trust the revised manuscript can now be accepted for publication in Nature Communications. Below is the point-by-point response to the criticism and suggestions.

Sincerely Yours,

Igor Musevic,

On behalf of Authors

Reviewer #1 (Remarks to the Author):

Comment: The present manuscript investigates in detail the continuous formation and annihilation of topological defects in a thin liquid crystal film in contact with a fluid during driven turbulent flow. The paper is very well written, concise and informative, and should be of interest to a wide soft matter community, also due its relation to active materials. The well-referenced introduction provides the explanations needed by non-specialists reading the paper. Detailed descriptions are provided of the experiment so that it could be reproduced easily by other groups. The discussion and interpretation of the presented results are reasonable and verified by the data and movies provided. Given the lately broad interest in topology, active materials and defect creation and annihilation, I believe that this manuscript would make a good contribution to Nature Comm. and I thus would recommend publication.

Reply: We would like to thank the referee for his/her appreciation of our work and recommendation to publish it in Nature Communications.

Comment: When reading the paper up to fig.2 the question arises which one of the defects is the +1 and which is the -1 defect. I do realise that this is explained later, but it might help the reader if this is already indicated in fig.2.

Reply: Thank you very much for this comment. We revised Figure 2 by adding labels, denoting which defects are +1 and which are -1.

Comment: During the annihilation process of the defect pair, is there an anisotropy in the defect dynamics, as it is seen in other experiments? And if so, is this comparable? A short discussion of this aspect might enrich the paper somewhat.

Reply: We thank the reviewer for this interesting comment. A number of experiments on defect interaction in nematics demonstrate the anisotropy of their dynamics during annihilation in the sense that + topological charge monopoles always move faster than the - monopoles. This is due to the different viscous drag force on - and + charge monopoles, which results in their different dynamics during attraction. All the reported experiments were performed for zero background velocity of the fluid, i.e. the liquid crystal was at rest and the velocity of each defect can be measured in laboratory frame, which is also at rest.

Figure 5b shows the measured distribution of defects' velocities in the laboratory frame of reference. It is similar to particles' velocity distribution, but much broader. Following your recommendation we made an analysis of the absolute magnitudes of + and - defects, separately, which are shown in SI Figure 7. One can clearly see that - defects on average move faster than the + defects during annihilation. This is contrary to reported behaviour, where the + defects were always faster. The discrepancy can be explained by the background velocity of LC, which was zero in all experiments reporting faster + defects. Namely, we see in our experiments that + defects are created always ahead of the - defects with respect to flow. During annihilation they are pulled back by their - neighbours, thereby slowing down, while the - defects are speeding up relative to laboratory system.

In order to measure relative velocity of both types of defects, the velocity of the background LC flow should be measured very accurately and at a large number of points. This is an extremely challenging experimental setting that, we believe, goes far beyond the scope of this manuscript. We commented anisotropy of defect dynamics at the end of Section 2.2. of the revised manuscript.

Comment: Also in fig 6(d) one could indicate which one of the defects is +1 and which is -1, because readers outside of the field may not be instantly familiar with the director configuration of both defects.

Reply: Thank you for this note. We changed former Figure 6(d), now Figure 4(p), by adding labels of +1 and -1 next to the defects, to clarify their configurations.

Reviewer #2 (Remarks to the Author):

Comment: The manuscript "Continuous generation of topological defects in a passively driven nematic liquid crystal" by Mur and Musevic depicts topological defect generation in dynamic flow in nematic liquid crystal confined in a hybrid anchoring condition. There are mainly two parts describing different experiments, i. e. 1) The dynamic structural transformation (explosion) of a hemispherical droplet of hybrid SmA/N state, and 2) defect formation in nematic thin-films upon passive dynamic flow (convection) driven by diffusion of small molecules. These two parts are linked, because the dynamics in the latter part was found by the observation of the former process. But, as far as described in the

manuscript, they are not very much correlated. As a result, the main point in the present manuscript became unclear, and thus in this context, the storyline needs improvement.

Reply: We agree that the story is split into two parts, which were not quite well connected in the original manuscript. Based on your opinion, we decided to move most of the story about "droplet explosion" to the Supplementary Information, leaving a very short part of it in the main text. We are convinced that reader's attention will be now strongly focused on the main story of the article, i.e. continuous defect formation.

Comment: As given in the title, the main topic of the present manuscript is the latter, i.e. the generation of the topological defects via the passive flow. However, this part is still too qualitative. With the aid of video analysis, the authors succeeded in describing the generation process of defects, together with the flow field distribution. Intuitively, the conclusion simply says, the interplay between the elasticity and the diffusion of the small PI molecules transferred from water phase to the N phase is the initial driving force of the flow, and non-equilibrium dynamics due to the flow-elasticity coupling is a role to generate such a unique defect formation state. But it looks almost all in the present manuscript.

Still it would be difficult for the readers to catch what was the newly-understood physics or more broadly, science brought by these experimental data and analyses. To describe more the physics, the authors may need more quantitative physical analysis as they admitted by mentioning "This means that their mutual elastic interaction gives rise to substantial dynamics of the defects. However, to say more about the elastic contribution to the flow dynamics of defects we need to know the flow velocity field of the LC in real time"

So, while I like the concept of the present work as well as high quality data, the manuscript in this stage is still too weak because of lack of new physics, and it is difficult to publish the present manuscript in Nature Communication.

Reply: We would like to thank the reviewer for critical assessment of our manuscript and recommendation to better understand the new physics in our experiments. We fully agree with the statement that "... it would be difficult for the readers to catch what was the newly-understood physics, or more broadly, science brought by these experimental data and analyses. To describe more the physics, the authors may need more quantitative physical analysis..."

We therefore made more quantitative analysis of new physics in our experiments by engaging the theory group and two new co-authors (Miha Ravnik and Ziga Kos). They made numerical simulations of our experiments, which are now presented in new Section 2.3. Numerical simulations. Briefly, they simulated the concentration-induced flows and continuous generation of topological defects by considering the dynamics of three mutually coupled fields: the velocity field describing the flow of the LC, the tensorial ordering field describing local order of LC, and the concentration field of the photoinitiator. Numerical simulations are now presented in new Figures 6, 7 and 8 and there is a

section in SI explaining numerical modelling with 3 videos of simulations. The simulations are in striking correspondence with the experiments shown in Figures 2 and 3 and with videos of the experiments.

The simulations show a transition from counter rotating vortices to turbulence at a certain value of Rayleigh number. While defects are never produced in the regime of counter rotating vortices at low Rayleigh number, where the flow is quite regular, they are regularly produced and annihilated in the regime of irregular, chaotic flow with high Rayleigh value. We found that the Ericksen number (Er) is the second parameter that describes the threshold for defect production. At low Er the tumbling deformation of the orientational field is counteracted by the nematic elasticity, while it fails to counterbalance it at high Er leading to tumbling and defect production. This is in clear correspondence with our experiments, which show exactly the same behaviour and is explained in good detail in new section 2.3.

Based on such a good correspondence between the theory and experiments, we were able to better understand the role of various forces that are driving the liquid crystal in a wedge geometry from laminar to chaotic regime and triggers defect production. This is discussed in good detail in new Discussion section and also new section in SI. We divide the flow-driving forces due to photoinitiator concentration in two classes: surface and bulk forces in wedge geometry, new Eq.1. We performed extensive numerical simulations by varying the sign and magnitude of individual surface force terms in Eq. 1. We observe that Marangoni force can induce irregular/chaotic flow and defect production only in case where higher concentration of photoinitiator increases the surface tension. We performed extensive measurements of the concentration dependence of surface tension of LC and water (described in SI) and in all cases higher concentration of the PI decreased the surface tension, which is quite common. This rules-out Marangoni surface force as a possible driving force to produce chaotic flow and generate defects. It is then clear that volume forces are likely to be the generator of defect production and several possible terms in the force volume field are identified that can do that. Most likely, the turbulence is triggered by a combination of several surface or bulk forces due to the diffusion and mixing dynamics in the concentration of the photoinitiator. However, this requires extensive new experiments that go well beyond the scope of this work. This is now discussed in detail in the revised manuscript.

Comment: 1. In this manuscript, still the relation between the surface condition and the flow dynamics are not clear. In addition, I wonder if the Marangoni effect at the interface is also playing for the process, or fully bulk diffusing flow? If so, is it possible to be further discussed with some kind of physical analysis, like surface tension measurement? But also it would be nice to mention more the diffusion and the internal flow.

Reply: We would like to thank the reviewer for this very important comment to elucidate the role of surface forces due to the gradients of the PI concentration field. To explore the role of Marangoni effect, we measured the concentration dependence of surface/interfacial tensions by setting up a new experimental method based on Laplace pressure difference measurements, as described in new SI. Briefly, by using a micropipette and controlled source of well-defined pressure, we inflated one liquid droplet either in air or another liquid. By increasing the pressure and measuring the inflated diameter

we can determine the surface/interfacial tension. We find that the surface tension of water decreases with increasing concentration of the photoinitiator. The surface tension of 8CB in the nematic phase also decreases with increasing concentration of the photoinitiator.

Figure 6. of the SI shows the schematic drawing of concentration distribution of the photoinitiator within the thin LC film, together with the values of surface tensions. At the 8CB-air interface, the surface tension of nematic 8CB slightly increases from 33 mN/m in thin to 35 mN/m in thick part of the wedge. On the other hand, the 8CB-water interfacial tension is much lower, i.e around 18 mN/m.

We performed extensive numerical simulations by varying the sign and magnitude of individual surface force terms in Eq. 1 and we observe that Marangoni force can induce irregular/chaotic flow and defect production only if the surface (or interfacial) tension increases with increasing concentration of photoinitiator. This is in clear disagreement with our surface tension measurements where in all cases surface tensions decreases with increasing concentration (see SI) which is common effect of surfactants.

Within the limitations of the presented minimal model and for the scanned parameter range (though quite extensive), the comparison between experiments and theory implies that Marangoni force is not likely to induce topological turbulence in our experiments. The second possibly relevant mechanism for the emergent material ow is the diffusion and mixing of photoinitiator within the bulk of the liquid crystal layer. This is now extensively discussed in revised manuscript.

Comment: 2. Related to 1, is there any method to make it possible to visualize the distribution of the PI concentration and surface tension?

Reply: We would like to thank you for this very interesting question. We considered optical fluorescence from the PI molecules as an elegant way of determining concentration field without no complex probes. To this aim we measured the fluorescence spectra of PI molecules and it has a peak at 510 nm when excited with UV light. Fluorescence from PI in water could be observed, but the PI fluorescent signal was very weak and it was not possible to measure the concentration of PI via the intensity of fluorescence. Interestingly, 8CB is also fluorescing with a peak at 440 nm and this fluorescent light is much stronger compared to fluorescence from PI dissolved in water.

Comment: 3. In the present case, is the use of 8CB having the N-SmA transition necessary for the induction of this defect generation phenomenon? I thought that the SmA core in the droplet is important for the explosion process, but it is not clear how it works for thin-film expansion and defect formation, especially in the case of the stripe experiment. Or can we use any Nematics for the same experiment?

Reply: Thank you for this interesting comment. Following your question, we made additional experiments with 5CB and 7CB liquid crystals. We observed the same continuous generation of topological defects in 7CB. 7CB exhibits only a nematic phase, and does not show a smectic phase, therefore we can definitely say that SmA-N transition is not a necessary condition for the observed generation of defects in the thin film. On the other hand, we made a number of experiments using 5CB,

but the defect generation did usually not appear. Only in one experiment we could see a slow dynamic behaviour in which a small number of defects were produced, quite possibly by a different mechanism. The key difference is that 7CB and 8CB are tumbling nematic liquid crystals, while 5CB is a flow-aligning one, The process of topological defect production in the tumbling-chaotic regime therefore strongly depends on the molecular structure of LC. We added a comment about this in the revised manuscript at the end of Discussion.

Reviewer #3 (Remarks to the Author):

Comment: This article titled "Continuous generation of topological defects in a passively driven nematic liquid crystal" investigated the continuous generation of topological defects with an integer topological charge. The approach of generating topological defects in liquid crystals (LCs) by the gradual diffusion of foreign substances at the water-LC interface is considered a counterpart for active nematic systems using biomaterials. The mechanisms and experimental methods for generating turbulence are new, and the qualitative analysis of the formation and behaviour of topological defects is well discussed. This study is expected to appeal to a broad readership of 'Nature Communications'. However, some parts need to be supplemented, and if the authors can discuss the contents of the following questions in the research results, the reviewer agrees to be published after the revision.

Reply: We would like to thank the reviewer very much for appreciating the novelty of our work.

Comment: Authors utilize "2-hydroxy-2-methylpropiophenone photoinitiator (PI)" as foreign material for generating turbulence. However, the author did not explain why this specific foreign material was used. Is there any reason to select it? Do other substances that can dissolve in water and diffuse into LCs show the same experimental results?

Reply: We found this effect by chance. We used 2-hydroxy-2-methylpropiophenone photoinitiator (PI) in some other experiments, where growth of smectic-A fibers was studied in water-CTAB surfactant solutions. Because the growth of smectic-A fibers is difficult to control, we tried to immobilize the LC fibre structure by adding PEGDA hydrogel (poly(ethylene glycol)-diacrylate) to water and 2-hydroxy-2-methylpropiophenone photoinitiator to polymerize PEGDA by UV illumination. During these attempts we noticed that 8CB is absorbing the PI from water, changes the smectic A phase into the nematic at room temperature and triggers generation of topological defects after the "droplet explosion".

We were searching for other materials that could initiate the same behaviour. We repeated the experiments with other substances that are poorly soluble in water and quite well soluble in the liquid crystal: oleic acid and acetone. As expected, they did have an effect on the LC phase, but in neither case the thin LC film formed. We also used sodium dodecyl sulfate (SDS) solution and a solution of a common

liquid hand soap, to reduce the surface tension of water, but again we were not able to see the phenomenon of spontaneous defect generation. We are convinced the material should be much better soluble in LC than in water and at the same time it should decrease the surface tension of water and LC to be able to trigger the defect generation. However, until now we could not find a replacement for the PI. We comment this in the revised manuscript at the end of Discussion.

Comment: From a chemical point of view, photoinitiators generate radicals when exposed to lights as a follow-up question. It is thought that this could also happen during optical microscopic observation, wouldn't this be an additional source of turbulence or 'explosion'? Please describe the presence or absence of radical formation or its effect.

Reply: Thank you very much for this comment. Indeed, we have considered this issue in our experiments, but did not comment on in the original manuscript. We did not observe any influence of the ambient or optical microscope illumination light on topological defect generation, the same process clearly took place also in dark room under red light illumination. It is known that 2-hydroxy-2-methylpropiophenone photoinitiator generates radicals by illuminating with 240 nm to 340 nm UV light. We observed no visual changes under illumination with light in optical microscope compared to red-light illumination. Motivated by your comment, we made additional experiments with PI. We measured the fluorescence spectra of PI and of PI in water. The PI was excited by UV light (peak at 370 nm, FWHM cca 20 nm) and it fluoresces in green, the peak of the emission is at 510 nm. When the PI in water was illuminated with 370 nm UV light under a microscope, we could see the generation of free radicals. Water became turbid and the LC film started spreading over water. This clearly shows that generation of free radicals under UV illumination has significant effect on surface tension and could influence our experiment. However, this process can be seen only under UV illumination and can be ignored in our experiments. We added some comments on this issue at the end of Discussion in the revised manuscript.

Comment: The authors stated that the flow of a fluid changes over time from laminar to turbulent. Is it possible to quantitatively explain (e.g., Reynolds number) when this transition occurs using the characteristics and thickness of the LCs?

Reply: Thank you very much for this question. Over time, the PI concentration in the liquid crystal film changes. This affects the flow velocity that increases with time and the flow field changes from laminar to turbulent. At the same time the dimensions of the film (thickness and width from the N/SmA to N/I interface) change. We estimated the Reynolds number by considering constant thickness (which in fact changes with time), measured at the time, when defects start forming. The Reynolds numbers we calculated were very low. We estimated that the defect formation started, when Reynolds number exceeded approximately $2 \cdot 10^{-5}$. This is an extraordinary low value for the onset of turbulence, however, similar value has been obtained for bacteria, $Re \sim 10^{-5}$ [H. H. Wensink, J. Dunkel, S. Heidenreich, K. Drescher, R. E. Goldstein, H. Lowen and J. M. Yeomans, Meso-scale turbulence in living fluids, Proc. Natl. Acad. Sci. 109, 14308 (2012)].

Comment: The temperature of the experiment affects both the diffusion rate of the photoinitiator and the phase transition temperature of LCs. It means that changing the experimental temperature can change the velocity and frequency of appearance of topological defects. Please verify this in the revised manuscript. And it might have the same function as controlling the velocity of bacteria in the active nematics systems; please discuss this.

Reply: We would like to thank you for this very interesting suggestion. All experiments were done at room temperature, which changed for a couple of degrees during the some months of duration of the experiments. Indeed, we expect that increase in temperature could substantially change the velocity and frequency of appearance of topological defects. We should stress that the velocity of the flow and the rate of generation of topological defects strongly increase with time already at room temperature. For example, the flow velocity itself typically changes for an order of magnitude during the duration of experiments, which is some minutes. By increasing the temperature this processes would most likely speed up even more. On the other hand, performing such an experiment at elevated temperatures is challenging from practical viewpoint. Because the water and LC surface have to be exposed to ambient air in our experiments, increasing the temperature would increase the rate of water evaporation, which would likely result in overall positional instability of thin LC film. Furthermore, convective flows could appear at increased temperature. We conclude that such an experiment is definitely of great interest, but would require significant redesign and testing of the new experimental set-up. This is probably the scope of future experimental work on this subject.

Comment: The authors analyzed the velocity and direction of turbulence in the LC by adding fluorescent particles. It might help readers understand if authors add vector fields with a colour map.

Reply: Thank you very much for this comment, which we fully considered and added velocities in places where it was measured.

Comment: The author explained the reason for using 8CB in the introduction, including references. If 5CB is used under the same experimental conditions, are not topological defects formed at the isotropic and nematic interface? If not, how does it behave?

Reply: Thank you very much for this question, which was also raised by Referee #2. In order to clarify this question, we conducted additional experiments with 5CB and 7CB liquid crystals. We observed the same continuous generation of topological defects in 7CB, which exhibits only a nematic phase, and does not show a smectic phase. We can definitely say that SmA-N transition is not a necessary condition for the observed generation of defects in the thin film. On the other hand, we made a number of experiments using 5CB, but the defect generation did usually not appear. Only in one experiment we could see a slow dynamic behaviour in which a small number of defects were produced, quite possibly by a different mechanism. The key difference is that 7CB and 8CB are tumbling nematic liquid crystals, while 5CB is a flow-aligning one. We added some comments about this in the revised manuscript.

Comment: This research can be classified as an LC research field. Thus, the abbreviation photoinitiator (PI) reminds many readers of polyimide. For this reason, it is recommended a different abbreviation.

Reply: Thank you very much for this important comment. Indeed, PI is the abbreviation for polyimide in LC community. To avoid confusion we replaced "PI" by "HMPP", which is a standard abbreviation for 2-hydroxy-2-methylpropiophenone photoinitiator, used in our experiments.

Comment: The author provided high-quality videos of topological defects. However, defects formed in the videos cannot be easily observed. So, it is recommended to add some markers in the video for topological defects.

Reply: Thank you very much for this note. We added some markers to the video to easily find the topological defects.

Comment: The article is well organized, but the same concept seems to be explained with too many duplicates in the manuscript.

Reply: Thank you very much for careful reading of our manuscript. The manuscript was heavily edited and we followed your advice to avoid duplicates.

List of changes:

1. We changed Figure 1, as some part of the text was moved to SI.
2. We changed Figure 2 by adding labels, denoting which defects are +1 and which are -1.
3. To reduce the number of figures, we made new Figure 3 by combining Figure 3 and 4 from the original manuscript.
4. We made new Figure 4 by combining former figures 5 and 6. Arrows were added to show the position of defects. Labels of +1 and -1 were added next to the defects.
5. We made new Figure 5 from previous Figure 7 by adding vectors showing velocity of tracer particles.
6. We added new Figures 6, 7 and 8 in revised manuscript.
7. Part of the text describing the “explosion” of LC droplet was moved to new SI.
8. We added new subsection 2.3. Numerical simulations with new Figures 6, 7 and 8.
9. Section 3. Discussion was heavily edited and expanded.
10. Subsection 4.4. Surface tension measurements was added to main text.
11. Main text was rephrased in various places, as indicated in the red-line version of revised manuscript. Some parts of the original text were deleted because they were repeated elsewhere in the manuscript.
12. New references 43, 44, 51 and 52 were added in the main text.
12. Supplementary Information was added with extensive description of surface tension measurements, analysis of defects' velocities and explanation of numerical methods.
13. Three new movies were added to SI.
14. PI was replaced by HMPP in all places it appeared.

REVIEWERS' COMMENTS

Reviewer #1 (Remarks to the Author):

The authors have addressed all my previous comments to full satisfaction, and obviously also a number of issues raised by other reviewers. I am happy with the changes made and recommend publication of this interesting paper.

Reviewer #2 (Remarks to the Author):

The manuscript was revised thoroughly and improved significantly. The newly-added theoretical explanation and simulation, and additional experiments such as surface tension measurements and observations enrich the scientific value of the work and make it more convincing. Most of my concerns were mentioned in in the present version of the manuscript. So the paper can be published in Nature Communications.

Reviewer #3 (Remarks to the Author):

The authors have conducted lots of efforts to improve their work, although one thing, variation-temperature experiments were not carried out because of the practical limit. The issue may be resolved in the future when the authors have the proper experimental set-up. So I recommend this work published as it is.

Response Letter to Reviewers' comments and criticism

We thank the reviewers for their fair and accurate assessment of our manuscript. Their constructive criticism and positive comments helped us to considerably improve the scientific quality of our work.

Sincerely Yours,

Igor Musevic,

On behalf of Authors